



# A portable Lightweight In Situ Analysis (LISA) box for ice and snow analysis

Helle Astrid Kjær[1], Lisa Lolk Hauge[1], Marius Simonsen[1], Zurine Yoldi[1], Iben Koldtoft[1], Maria Hörholdt[2], Johannes Freitag[2], Sepp Kipfstuhl[1,2], Anders Svensson[1], and Paul Vallelonga[1,3]

1 Physics of Ice, Climate and Earth (PICE), Niels Bohr Institute, University of Copenhagen, Copenhagen 2200, Denmark
2 Alfred-Wegener-Institut Helmholtz-Zentrum für Polar- und Meeresforschung, Bremerhaven, Germany
3 UWA Oceans Institute, University of Western Australia, Crawley, WA, Australia

**Correspondence:** Helle Astrid Kjær (hellek@fys.ku.dk)

10 **Abstract.** Polar researchers spend enormous costs transporting snow and ice samples to home laboratories for "simple" analyses in order to constrain annual layer thicknesses and identifying accumulation rates of specific sites. It is well known that depositional noise, incurred from wind drifts, seasonally-biased deposition, melt layers and more, can influence individual snow and firn records and that multiple cores are required to produce statistically robust time series. Thus at many sites core samples are measured in the field for densification, but the annual accumulation and the content of chemical impurities are 15 often represented by just one core to reduce transport costs. We have developed a portable Light weight in Situ Analysis (LISA) box for ice, firn and snow analysis capable of constraining annual layers through the continuous flow analysis of melt water conductivity and peroxide under field conditions. The box can run using a small gasoline-generator and weighs less than 50 kg. The LISA box was tested under field conditions at the deep ice core drilling site EastGRIP in Northern Greenland. Analysis of the top 2 metres of snow from 7 sites in Northern Greenland (Figure 1) allowed the reconstruction of regional 20 snow accumulation patterns for the period 2015-2019.

## 1 Introduction

To evaluate future sea level changes surface mass balance (SMB) determinations of the major ice sheets and ice caps is an important constrain. Theoretical predictions on the change in SMB is made based on advanced ice sheet models, but rely on accurate accumulation estimates from also past periods(Montgomery et al., 2018). Accumulation of the past can be 25 reconstructed by means of ice and firn core analysis. To estimate ice core ages chemical profiles are often used to generate annually resolved timescales(Svensson et al., 2008; Winstrup et al., 2012), however, those are spatially limited for practical reasons; drilling campaigns of deep ice cores are expensive as they require the maintenance of large scale facilities for several years, while drilling campaigns of shorter snow and firn cores can be done cheaper, though still costly in terms of time and money.





Having retrieved such snow/firn cores annual layers and accumulation rates can be directly reconstructed by investigating the chemical impurities within the ice. Several of these impurities have annual cycles in Greenland snow and ice; eg. insoluble dust particles and calcium peak in early spring as a result of increased storm activity, acids peak in spring time following the Arctic haze phenomenon and higher signals are observed with acid from volcanic eruptions, peroxide is driven by light

processes and peak in midsummer(Gfeller et al., 2014; Legrand and Mayewski, 1997). Having one or more profiles of an ice core site, an accurate time scale can be established by dating and, if in addition the density is measured, the local accumulation rate of a site can be reconstructed(Philippe et al., 2016; Winstrup et al., 2019).

The chemical profiles in ice cores can be obtained by means of Continuous Flow Analysis (CFA), a fast way to obtain high resolution chemical profiles, performed in home laboratories(Bigler et al., 2011; Dallmayr et al., 2016; Kaufmann et al., 2008).

A generic CFA consists of a melt head, which splits the sample stream into an inner uncontaminated line and an outer possibly contaminated line. The inner line is then split further allowing a small amount of sample water for each analytical measurement. Field campaigns with in-camp melting and CFA is not done on routine basis, as the analytical setups require space, time and warm laboratories on site(Schüpbach et al., 2018). These are limited to larger ice core drilling stations in Greenland(NEEM community Members et al., 2013) and Antarctica where the heavy and delicate instruments can be flown in and out by large

cargo aircraft and warm buildings or areas exists with space enough to build up a laboratory.

Here we have developed a truly field portable Continuous Flow Analysis (CFA) setup that can determine annual layers in snow, and firn. We made a system optimized for deep field deployment, which can be powered by a small gasoline generator and can be operated in below-zero conditions. In addition, we made it light weight enough to be transported in the field by two persons. As time in the deep field is often limited we also aimed for a device that provides results quickly such that the spatial

variability at a site can be investigated within a day of field work and also an instrument that can be used by non-professionals. In the summers of 2017 and 2019, we demonstrated the LISA box under Greenland field conditions at the EastGRIP ice core drilling site (see map Figure 1) by reconstructing accumulation from snow cores covering the top two metres of snowpack. Although the LISA box may be used to melt ice cores, we will focus on the application to snow and firn.



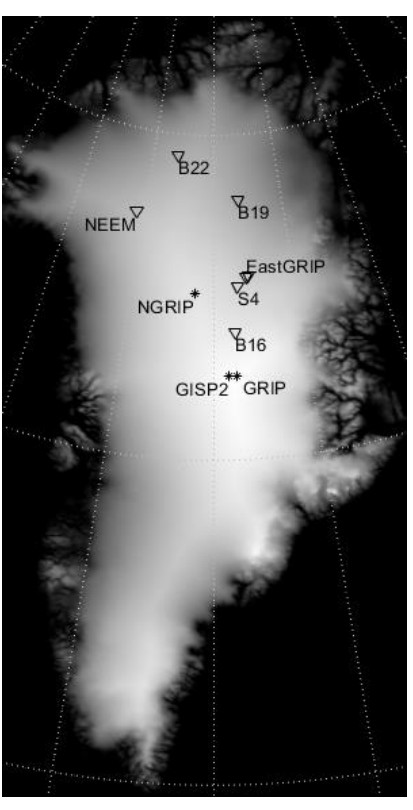

**Figure 1: Position of the sites studied using the LISA box (triangles). Stars indicate deep ice core drilling sites. Note that at EastGRIP two sites (S5 and S7) were investigated. Background altitude map of Greenland is based on the SeaRISE dataset(Bamber, 2001; Bamber et al., 2001).**

## 2 Materials and methods

The LISA box is a small Continuous Flow Analysis system set up to melt snow, firn and ice cores and analyze the melt water for conductivity and peroxide.

Overall it consists of a square foam box with a lid that insulates from outer conditions. The outer dimensions of the box are 0.585 m high, 0.475 m wide and 0.765 m long. The foam is 10 cm thick making the inner dimensions 0.35m x 0.35 x 0.58 m, respectively. In the lid is placed a circular melt head and inside the box, which is temperature controlled, a small version of a continuous flow analysis system is set up including a miniature computer for instrument control and storage (see Figure 2).

Some additional components are required outside the LISA box. These consist of a handheld generator for power and a wastewater container and pump is also placed outside the LISA box, as well as a screen and keyboard for easy instrument adjustments. Below the LISA box is presented in more detail.



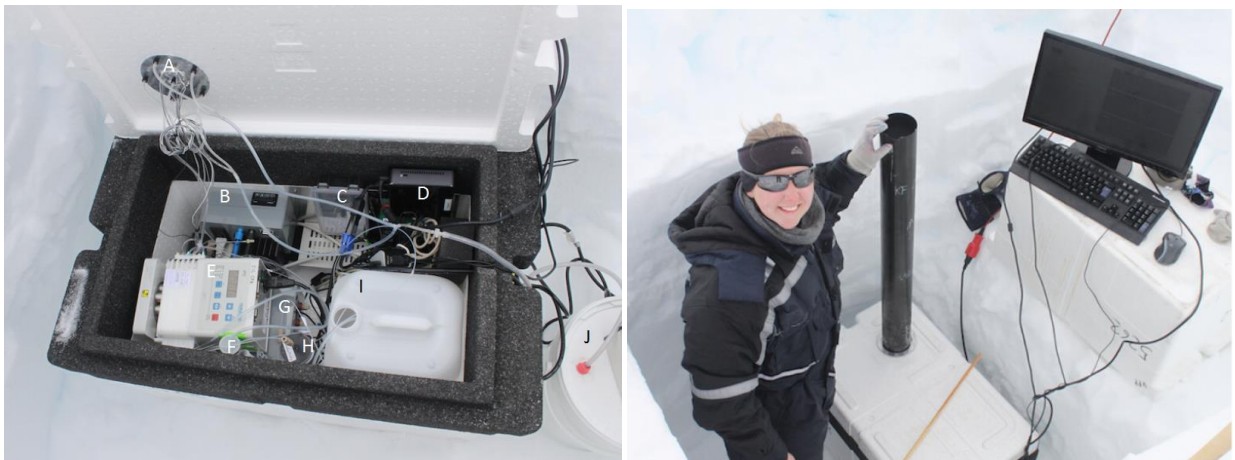

**Figure 2: Left) An inside look at the LISA box; A) the underside of the melt head, B) melt head temperature regulator, C) temperature PID that stabilizes the inner temperature of the box, D) computer, E) peristaltic pump, F) debubbler, G) fluorescence box (F-box), H) conductivity metre and I) chemical wastewater container, J) container for collecting waste water from the outer section of the melt head located outside the LISA box. Right) LISA box during field work 2017. The snow is collected in the black liner and melted continuously, while data can be observed online on the attached screen.**

## 2.1 Melt head and melt speed registration

The melt head (Figure 2-A and in supplementary material Figure S3) used to melt the snow, firn or ice sample, is placed in the lid of the foam box, reaching both the inner and outer side of the foam box with the intention that the sample is melted on the outside, while melt water is transported inside the box for further analysis.

The melt head is circular to allow for analysis of full size snow and ice cores to avoid time-consuming cutting in the field. The melt head has an inclined surface to divert the outer part of the melted core to waste, and an inner inclined conical surface for collection of the sample melt water (supplementary material Figure S3). The inclination prevents bubbles from entering the sample line, as the conical shape melts the different depths of the firn core at the same time and "floods" the lower part of the inner cone. The volume of the inner cone determines how much mixing happens in the sample stream before it is distributed to the analysis unit. As is done for all CFA systems to prevent contamination for the outer part of the core the inner melt head cone melts more water than is being drained, thus producing an overflow of inner meltwater out toward the waste drainage.

The melt head is heated by heat cartridges (HHP30, Mickenhagen GmbH, Lüdenscheid, Germany), which are controlled by a Proportional Integral Derivative (PID, CN7532, Omega Engineering) device (Figure 2-B, Figure S1, supporting information). One of the heat cartridges has an inbuilt J-type thermocouple which provides the temperature for the PID, so that the temperature of the melt head can be regulated to obtain an optimum melt speed. The optimum melt speed is site dependent and can thus be varied with the density of the sample and the expected annual layer thickness.

The melt head is made of solid aluminium chosen for its low density combined with high thermal conductivity, and excellent corrosion resistance, with an inner diameter of 11 cm to fit a 10 cm snow core. The inner cone is 7 cm higher than





the deepest point of the melt head, which is 6 cm lower than the outer rim. There is one drainage hole in the centre of the melt head and 5 outer drainage holes. The inner cone volume is 3.8 cm$^3$. Additional specifications of the melt head can be found in supplementary Table S1.

The melt speed is a crucial parameter to obtain as it is used to reconstruct the depth of the analysed sample. Several ways exists to determine melt speed in CFA systems, including encoders, lasers and image recognition(Bigler et al., 2011; Dallmayr et al., 2016; McConnell et al., 2002). The melt speed in the portable CFA developed here is simply registered by simultaneously determining the amount of core left above the melt head with a ruler and the time difference required to melt 3 cm of ice. While this method is clearly imprecise, compared to the more advanced options, it is also the simplest as it has no complicated electronics necessary and require a minimum of weight as the only tool necessary is a ruler, chronometer and a notebook.

## 10  2.2 Temperature stabilizer of inner box

When aiming to move a Continuous Flow Analysis to the field it is crucial to ensure a stable temperature environment for the measurement apparatus and chemicals inside the box. Thus inside the LISA box is installed a custom built thermostat to ensure a stable temperature inside the box of between 17 ℃ and 18 ℃ (Figure 2-C). The thermostat and its digital display is mounted on a stainless steel plate that is fixed on the box wall. The thermostat is connected to 200W compact fan heater (FCH-FGC1 15  series, Omega Engineering) that sits at the bottom of the LISA box, guarded with a fence to avoid contact with the other box contents. The fan is always active to ensure air is well mixed within the box and to distribute heat when required.

## 2.3 Computer and communication devices

To control measurement devices, melt head temperature and to save data a miniature computer is mounted on the wall of the box (Figure 2-D), connected to an USB hub and a RS232-to-USB converter (Digi International) also inside the box, as well as 20  to a screen and keyboard outside the box.

The wires for mouse and keyboard positioned outside the box goes at a small triangle cut at the side of the LISA box, as does the screen cable. The screen is connected through a series of adapters (VGA-HDMI and HDMI-HDMI).

The program that runs the CFA inside the LISA box is a custom built Labview program with an interface showing the measurements in real time, so that a first assessment of the data can be done while in the field allowing for a first data quality 25  assessment already while measuring.

## 2.4 Continuous flow scheme

The overall Continous Flow Analysis set up for the LISA box is shown in Figure 3 and builds on conventional CFA used for the analysis of ice cores using PEEK connectors and PFA (1.55 mm i.d., 1/8" o.d., IDEX) tubing (Bigler et al., 2011; Kaufmann et al., 2008).



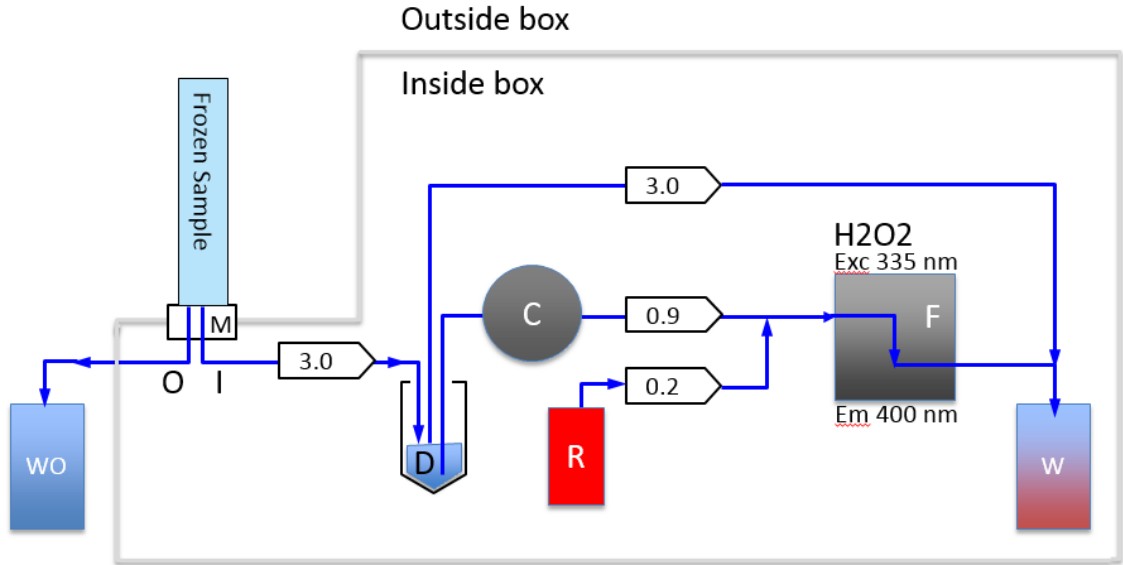

**Figure 3: The Continuous Flow Analysis (CFA) setup for the LISA box. The frozen sample is melted on the melt head (M) and the meltwater is split in an inner uncontaminated sample line (I) and outer possibly contaminated lines (O). The outer line is collected in a waste bucket outside the box (WO). The inner line is pumped to a debubbler (D), which is kept at a fixed volume by an overflow line. From the debubbler the sample reaches the conductivity metre (C ) after which it is mixed with reagent (R) prior to analysis by fluorescence (F). The mixed water is afterwards collected in a waste bucket (W). Active pumping is illustrated with pentagons and numbers within indicate the flow rate in mL/min.**

### 2.4.1 Debubbler

An Ismatec IPC 8 channel peristaltic pump and Tygon pump tubing are used for the inner line sample transport through the

CFA. A debubbler (see Figure 2-F, Figure 3-D) is implemented early in the system to ensure that potential air pumped into the

system or short pauses in melting do not interfer with the measurements downstream. The debubbler is a small acuvette with

one inlet line and two outlet ones. The inner melt water part from the melt head is pumped to the debubbler with a speed of 5

mL/min. The volume in the debubbler is controlled by one outlet line positioned at a fixed height in the debubbler actively

pumping excess melt water/air to waste (5 ml/min) to ensure constant volume in the debubbler. The second outlet line is the

main sample line and is positioned deep in the lower part of the debubbler. It is pumped with a speed of 1 mL/min (or 2 mL/min)

depending on setup) to further analysis.

### 2.4.2 Conductivity meter

The water is led from the debubbler to an AmberScience 3082 conductivity metre(Bigler et al., 2011) (Figure 2-H, Figure 3-

C). The response time (5-95% of signal) of the instrument itself is just 3 seconds. But typical measurement resolution is only

on the order of 20 seconds as a result of upstream smoothing in the debubbler and melt head.





### 2.4.3 Fluorescence measurements-H2O2

From the conductivity meter the sample is mixed with a reagent and measured by means of fluorescence. A custom box for fluorescence analysis was built for this purpose to limit the space used within the LISA box (F-box, Figure 2-G, Figure 3-F). The F-box allows for the determination of two chemical species by means of fluorescence and is powered by a 5 V power

supply.  The two fluorescence systems in the F-box can be independently activated or deactivated using two switches at the front of the F-box. Serial connectors and RS232 communication protocols were adopted to ensure stability also when moving the box between sites in the field.

Inside the F-box each of the two analytical lines consist of a 10 mm path length cuvette (176.766-QS, Hellma) and a photomultiplier tube (PMT 9111, Sens-Tech, UK) in a light-tight aluminum holder with a specific optical filter in front

depending on the desired frequency of outgoing light(Bigler et al., 2011).

In the field we only tested the F-box for the determination of peroxide ($H_2O_2$), which is known to show clear annual signals in Greenland snow(Frey et al., 2006; Sigg and Neftel, 1988). The reagent is kept in a bottle at the bottom of the LISA box and is added to the melt water sample by 0.14 mL/min. The response time (5-95% of signal) on the peroxide system is just 13 seconds if fully optimized (Röthlisberger et al., 2000), but similarly to the conductivity it is influenced by upstream smoothing and

resulting response is on the order of ~40 sec. A LED of 335 nm was used to excite the sample and detection was done at 400 nm. The $H_2O_2$ reagent consists of 1L purified water, 0.61 g 4-ethylphenol, 5 mg peroxidase type II, 6.18 g $H_3BO_3$, 7.46 g KCl, 150 μL NaOH (30%)(Kaufmann et al., 2008; Röthlisberger et al., 2000). The $H_2O_2$ reagent was prepared prior to entering the field (up to 1 month in advance) and kept frozen until use.

In addition to the $H_2O_2$ analysis system the F-box is prepared for the determination of calcium. The LED for the calcium is

340 nm and the emission is determined at 495 nm. The reagent used is mixed with 1 mL/min sample at a speed of 0.14 mL/min and consist of 750 mL purified water, 20 mg QUIN-2 Potassium hydrate, 2.91 g PIPES and buffered to pH7 by 1 to 1.5mL NaOH (30%)(Kaufmann et al., 2008; Röthlisberger et al., 2000). We have successfully used the calcium reagent under normal laboratory conditions (Bigler et al., 2011) after it was frozen for months and thus also the calcium should be an option for analysis in the field, though it has not been explicitly validated there.

The mixture of reagent and sample water is collected in a 10 L waste bucket inside the LISA box (Figure 2-I) so that it can be brought back from the field for proper chemical waste handling. With a combined flow of just 2.2 mL/min when analyzing both conductivity, peroxide and calcium and a melt speed on the order of 3 cm/min, the waste container will only need replacement after analyzing approximate 130 m of ice.

### 2.5 Vacuum pump

The 5 outer lines from the melt head are each 1/3 inch wide (Tygon R3603), but are inside the box connected to just one (½ inch) line. This line exits from a small hole in the side of the LISA box. Here it is tightly connected to the lid of a 5L bucket (Figure 2-J). To the lid of the bucket is also connected a tube, which in the other end is attached to a vacuum pump (VWR,



PM27330-84.0, Pmax 0.3 bar, 65W) to ensure that the bucket constantly has a vacuum that works to quickly remove all wastewater from the melt head. During melting this vacuum will build up and one can adjust the strength of pull by letting some air into the bucket by releasing the lid. As this water is just melted snow and thus uncontaminated it can be dumped at site when full or collected for further analysis in home laboratories as wanted. The vacuum pump is powered from the same
electrical supply as the LISA box.

## 3  Field work

The LISA box was tested at the EastGRIP deep ice core drilling camp (75.63N, 36.00W, 2708 m a.s.l.) located near the onset of the North-East Greenland ice stream.

### 3.1 Field work in 2017

In 2017 the first set of field experiments were conducted using the LISA box under freezing conditions (-20 ℃) outside at EastGRIP camp (see Figure 2). The total weight of equipment brought to the field in 2017 amounted to 78 kg, split in three boxes; the LISA box, and two zarges boxes that included several spare parts.

The LISA box was set up to only measure conductivity. We succeeded in melting several shallow snow cores from surface to about one meter depth in conditions of -20°C and 5 m/s wind by positioning the box in a snow pit and using a small generator
for energy provision. The melt head temperature was set to between +35 and +45 C. Snow cores were collected using carbon fibre liners of 1 m length, 10 cm inner diameter and 1 mm wall thickness (called "liners") (Schaller et al., 2016). The snow cores were melted directly from the liners by holding the liner slightly above the melt head. Melt speed was registered by measuring the distance from the top of liner down to the top of the snow surface inside the liner about every three cm.  Two persons operated the box, one measuring the melt rate and the other supporting the liner and evaluating the online results.
Figure S3 (supplementary material) shows the conductivity results obtained in 2017. This test confirms that the LISA box can work in freezing outside conditions.

### 3.2 Field work in 2019

In 2019 we brought a total of 63 kg to the EastGRIP site, split in two boxes; the LISA box and one zarges box containing spare parts. Both peroxide and conductivity were determined in several snow cores. When assembled the LISA box weighted 50 kg
excluding generator. The box was operated at +20°C inside the main building at the EastGRIP site. Samples were collected in liners from 7 sites in Northern Greenland in May and July 2019 (Positions given in Table 1, map Figure 1). The liners were stored frozen for up to two month prior to analysis. Most samples were analyzed directly from the liners similar to the 2017 sampling technique, but some samples were transferred to plastic bags for storage prior to analysis. Depth registration was done every approximately 100 seconds and melt speed varied between 2.3 cm/min and 3 cm/min. Two persons operated the





**Table 1: Accumulation as determined by the summer peroxide peak measured by means of the LISA box for several sites in Northern Greenland. Densities are estimated based on 1 metre snow core weights. Also shown is accumulation from other sources a: Kjær et al (2020) (~20 yrs), b: Vallelonga et al. (2014) (50 yrs), c: Rasmussen et al (2013) (~3000 yr), d: Masson-Delmotte (2015) (~300 yrs), e: Schaller et al (2016) (3-4 yrs) , f: Weissbach et al. (2016) (~500 yrs), g: Karlsson et al (2020) (300 yrs), h: Nakasawa et al (2020) (7 yrs), i: Kuramoto et al, 2011.**

| Site | Position | Density (Kg/m3) | | Water eq. accumulation (cm) | | | | | | |
|---|---|---|---|---|---|---|---|---|---|---|
| | | | | this work | | | | | | others |
| | Latitude-N Longitude-W | 0-1m | 1-2m | 2017-2018 | 2016-2017 | 2016-2015 | 2015-2014 | mean | std | |
| B22 | 79°18'35.6" 45°40'26.3" | 326 | 357 | 12.52 | 18.84 | 12.25 | 15.04 | 14.67 | 2.64 | 14.5[f] |
| B19 | 77°59'33.4" 36°23'32.0' | 332 | 352 | 10.89 | 11.96 | 13.28 | 16.01 | 13.03 | 1.91 | 9.4[f] |
| | | 342 | 354 | 10.51 | 13.72 | 12.5 | 19.39 | 14.03 | 3.3 | |
| NEEM | 77°27' 51°3.6' | 346 | | 16.44 | - | - | - | 16.11 | | 23.5[a],20.3[c], |
| | | 350 | | 19.95 | - | - | - | 19.55 | | 20.3[d], |
| | | 352 | | 18.51 | - | - | - | 18.14 | | 22.5[e],17.6[i] |
| B16 | 73°56'07.9" 37°36'58.2" | 346 | 358 | 18.99 | 16.86 | 16.84 | - | 17.56 | 1.01 | 14.1[f] |
| S7-EastGRIP (ice stream) | 75°37'44.4'' 35°58'49.3'' | 378 | 418 | 16.75 | 16.62 | 19.48 | - | 17.62 | 1.32 | 11.2[b], 13[g,] 14.9[h], 14.5[h] |
| S5 EastGRIP (shear margin) | 75°33.296' 35°37.377' | 347 | | 16.13 | - | - | - | 16.13 | | 14.6[a] 14.0[e] |
| | | 332 | 367 | 16.97 | 17.98 | 15.39 | - | 16.78 | 1.07 | |
| S4 (50 km upstream from EastGRIP) | 75°16.236' 37°00.444' | 346 | 367 | 17.48 | 15.56 | 12.56 | 13.84 | 14.86 | 1.85 | |

box, one supporting the liner and measuring the amount of snow left during melting and the other person evaluating the online results and checking flowing of water in the system in the LISA box.

A total of 18 metres of snow was analyzed for conductivity (Figure 4) and peroxide (Figure 5) over 6 measurement days. Each

5   100 cm snow core was weighed to a precision of 10g and using the inner diameter of the liner of 10 cm the density for each 1 meter section was determined with an uncertainty from the scale <3 kg m$^{-3}$. In addition the cores from B16, B19 and B22 sites high resolution densities were obtained by means of computed tomography(Freitag et al., 2013; Schaller et al., 2016). It is worth noting that the two operators handling the LISA box during the 2019 field season had little previous CFA experience and thus the successful measurements obtained demonstrate that the LISA box can be handled also by non-experts.

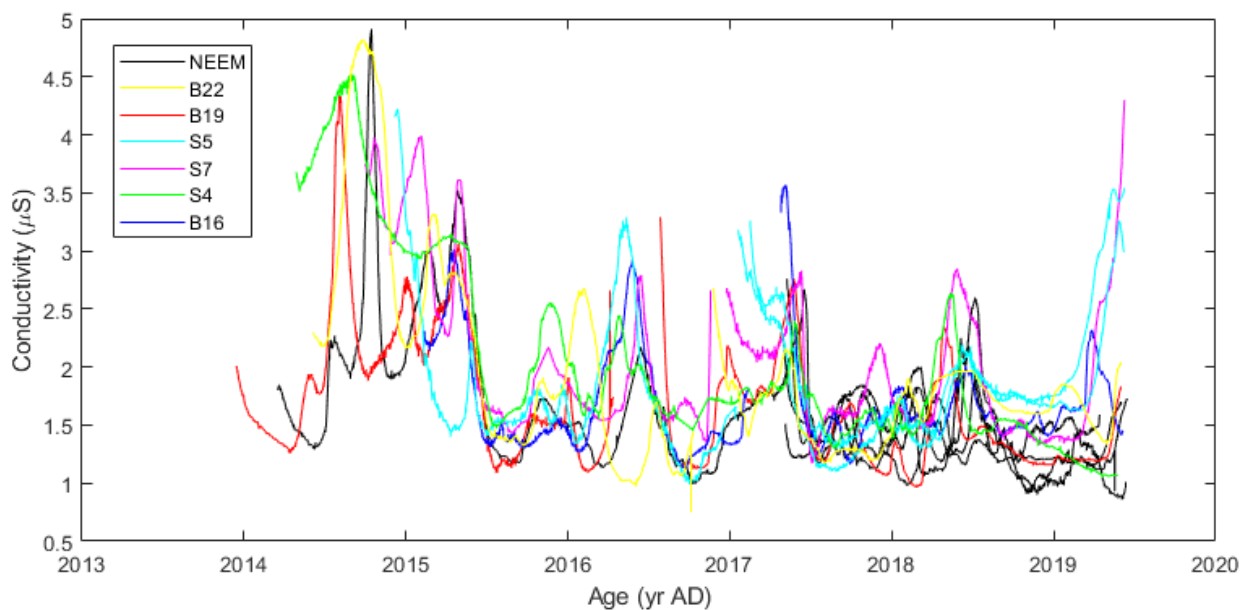

**Figure 4: Conductivity for 7 sites in North Greenland on an age scale. Note the high values centered in winter 2015 likely reflecting the eruption of the Holuhraun vent of Bárðarbunga volcano in Iceland (29 August 2014 to 27 February 2015)(Du et al., 2019b; Gauthier et al., 2016; Schmidt et al., 2015).**

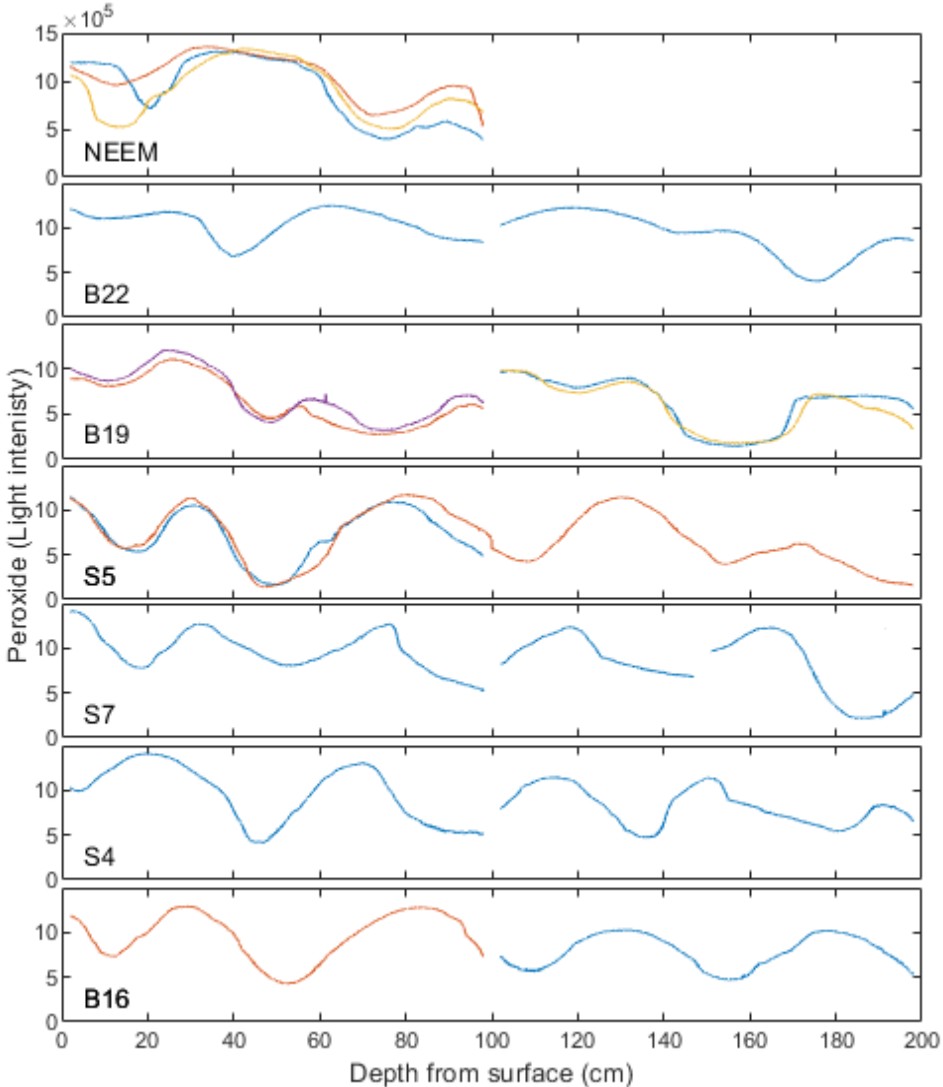

**Figure 5: Peroxide with depth for 7 sites in North Greenland (clockwise from North-West). For the sites NEEM, B19 and S5 several cores were retrieved. Colors indicate when multiple snow cores were collected or when the top and bottom of a pit was not measured in one run. The data are not calibrated and the y-axis is raw light intensities and thus while the level reflects the peroxide concentration absolute values are not to be trusted.**

## 4 Results

The peroxide results obtained from the LISA box in the 2019 field season are shown in Figure 5. Despite the data being uncalibrated, we observe in the raw light intensity counts annual cycles of summer high peroxide and winter minimum as expected (Frey et al., 2006; Sigg and Neftel, 1988). It is worth noticing that at the B19 and S5 site, where the snow cores were retrieved just few centimeters apart the resemblance between the measurements demonstrates the repeatability of the LISA





box. At NEEM samples were taken 10 m apart, and while one can observe the same features, the variability between the three
sets of measurements are larger, and also in the conductivity at the EastGRIP site (2017 field season) the two snow cores from
the same snow pit show significant variability, reflecting that the typical wind driven surface features operate on metre rather
than centimetre scale and confirming the need to take multiple cores from a site with metre intervals if the aim is to know a
sites mean chemical deposition (Gfeller et al., 2014; Laepple et al., 2016; Schaller et al., 2016).

An age scale was determined by linearly interpolating between $H_2O_2$ peaks by assuming the peak peroxide occurs in mid-summer (Frey et al., 2006; Sigg and Neftel, 1988). In addition, the bottom age was determined using the mean annual layer
thickness from all the peroxide peaks at that site relative to the amount of cm since last peroxide peak. Thus generating this
timescale, we are assuming equal accumulation throughout the year as is often done when dating ice cores based on just one
proxy. In general, the 2 m core sections reach back 4-5 years dependent on the site accumulation.

Using the H2O2 based dating we observe in the conductivity (Figure 4) as expected an annual peak centered around spring,
but very wide, and often with multiple events within. This suggest either 1) that our assumption that annual layers can be dated
using just the peroxide summer peaks in central Greenland snow is inaccurate. But while low accumulation sites or sites with
melt layers can influence the photolysis in the snow pack and thus the $H_2O_2$ stored within (Frey et al., 2006; Sigg and Neftel,
1988), here we are dealing with relative high and dry accumulation, thus we are confident that using $H_2O_2$ for annual layer
recognition is a reasonable approach. 2) that using conductivity in spring for dating snow and ice cores is not the best option
for timescale reconstruction in Northern Greenland surface snow or 3) that the seasonal timing of conductivity deposition over
Greenland varies. This is not unexpected as the conductivity in snow and ice, while in Greenland mainly driven by $H^+$ can be
highly influenced also by forest fire acids and or salt content. We note also that from 14th August 2014 until 15th February
2015 the VEI 1 eruption of the Holuhraun vent of Bardarbunga volcano occurred (Du et al., 2019b, 2019a; Gauthier et al.,
2016; Schmidt et al., 2015) and we find enhanced conductivity reaching almost 5 µS between June 2014 and May 2015
resembling the Holohraun eruption (Figure 4). This suggests that our dating is slightly off, but generally in agreement with the
expected time period. This is acceptable when considering the dating of the core is assuming constant accumulation throughout
the year and proofs that the box can be used also for identifying volcanic reference horizons often used to cross date individual
ice cores if these are not all annual layer counted. Below we discuss further the results from the LISA box, when applied to
reconstruct accumulation in Northern Greenland.

## 4. 1 Reconstructing spatial variability of accumulation in Northern Greenland.

Only a few accumulation estimates exist from Northern Greenland and most are older studies from the mid 90's (Montgomery
et al., 2018; Schaller et al., 2016; Weissbach et al., 2016). By combining the age scale based on the peroxide with the obtained
one metre density estimates, annual estimates of water equivalent accumulation were derived (Table 1, Figure S4-suplementary
material) for the 7 sites in Northern Greenland. When the years overlapped between the first and second meter the density used
was a combination of the top and bottom estimates based on the relative amount of the year falling into each. We note that the



meter averaged densities used in this study in the annual resolution in mean overestimated the accumulation by 4.7 % compared to the high resolution snow liner densities performed at B16, B19 and B22 by means of portable computer tomography (Freitag et al., 2013). Also shown in Table 1 are earlier accumulation estimates from the sites when available (Masson-Delmotte et al., 2015; Rasmussen et al., n.d.; Schaller et al., 2016; Vallelonga et al., 2014; Weissbach et al., 2016).

We do not observe any consistent increase or decrease for all sites compared to earlier estimates and thus no signature of speculated Clausius–Clapeyron (CC) relation of increased accumulation with temperature. The B22 site in Northernmost Greenland is within uncertainties equivalent with earlier estimates, while for the B19 site we find higher accumulation (Weissbach et al., 2016). For NEEM further to the west we find much lower accumulation compared to earlier estimates. We note that our reconstructed accumulation at NEEM rely on just one year (summer 2018-summer 2019), which is known to be

a year of low accumulation in Northern Greenland, while earlier estimates rely on several hundreds of years (Masson-Delmotte et al., 2015; Rasmussen et al., 2013). For the site B16 and the sites closer to EastGRIP (S5 and S7) our results from 2019 (~14.9-17.6 cm w.eq annually) are considerably higher than the 11.2 cm w.eq previously observed in the 400 yr NEGIS ice core obtained at the EastGRIP site in 2012 (Vallelonga et al., 2014), and also larger than the more recent estimates from radar (13 cm w. eq, (Karlsson et al., 2020)), firn cores (13.7 and 14.6 (Kjær et al., 2021)),  snow pits (14.6 cm w. eq.(Nakazawa et

al., 2020)) and the 14.1 cm w.eq observed at B16 site (Weissbach et al., 2016). Our results from EastGRIP 2017 using the conductivity (see supplementary) is more consistent with previous results.

We note that the years 2016-2018 were observed to have high North Atlantic Oscillation (NAO) and Arctic Oscillation (AO) winter indexes known to enhance winter precipitation in the northeast, and be anticorrelated with precipitation in the northwest (Koyama and Stroeve, 2019). Further the years analysed here also had high summer accumulation based on satellite estimates

in the North-Eastern area. Thus our generally higher accumulation east of the ice divide compared to previous estimates of accumulation can be explained by the atmospheric settings, which simultaneous explains the low estimates at NEEM. We also highlightthat variability between individual years for eg. B22 and S4 is larger than the difference to previous estimates from the area, and that at the sites where more snow cores were obtained (EastGRIP-2017, NEEM, S5 and B19) the difference in accumulation between individual years is up to 20% (Figure S4, Supplementary information) further other studies have shown

a similar variability investigating shorter firn cores (NEEM-25%, EastGRIP -30%(Kjær et al., 2021)). This illustrates the need to do several measurements over time at each site if aiming to reconstruct the climatic mean accumulation or to use firn cores to do so. We note that using the high resolution CT scanning densities instead of the 1 metre mean densities would not alter the above observations.

## 5 Lessons learned and future improvements of the LISA box

While, we have shown results from a first version of a LISA box, we will continue the development for a number of reasons. In this sections we discuss some of the lessons learned and suggest future improvements.



During field work in 2019 by accident the box internal flow was not completely emptied for water prior to storing. Later stored at freezing conditions, the expansion of the water during freezing resulted in a broken flow cell in the F-box. While this can be avoided simply by ensuring the box is empty for water when ending analysis, we will look into the optimization of the entire flow design, to ensure that such damage can be avoided or at least easily diagnosed.

During melting of the snow cores percolation up to 3 cm of water into the snow column above the melt head was observed. This causes additional smoothing and depth uncertainty to the data obtained by the LISA box. The melt head design can be improved to reduce percolation and ease the overflow decontamination procedure, for example by having extra drain holes and slits or channels etched into the melt head surface radiating outward from the center, both in the inner cone and the outer waste surface. These slits will provide an "escape" path for the liquid meltwater instead of being sucked in by air pockets in

the porous firn that is yet to be melted.

Further, the main uncertainty on the reconstructed accumulation from analysis with the LISA box is a result of uncertainties in the depth registration. One could instead use image recognition in combination with a camera to identify the melt speed as is done in the Desert Research Institute (DRI) Reno CFA or a continuous laser distance for the depth registration as done in other CFA laboratories (Dallmayr et al., 2016), however such systems will complicate the box and make it less practical to use

for non-experts. The camera pattern recognition is further hampered by the percolation of melt water, but would be the better option in case LISA were to be used on ice samples rather than snow. With a more stable frame perhaps a laser depth registration system could be implemented.

Finally, we acknowledge that basing the annual layers on more than just one chemical species would increase the accuracy of accumulation estimates and could diminish the overall variability in the resulting data. As the LISA box has the option of two

fluorescence lines as stands, it would be easy to add also either Calcium or Ammonium CFA methods(Bigler et al., 2011; Röthlisberger et al., 2000). Another addition could be insoluble dust by means of Abakus as is also often done in CFA.

While we have proven the LISA box in freezing, but relatively dry Greenland conditions, it remains to be tested in other climatic conditions, such as in high altitude glaciers, and colder Antarctic conditions. As the box is temperature regulated, and if enough energy is delivered to keep the box running at set temperatures, the cold temperatures in Antarctica should not in

principle cause additional complications. The energy consumption may however increase thus making the operation heavier in terms of gasoline for the generator, and one may consider moving the outside pump used to generate vacuum and subsequent waste container into the warmth of the insulated box, despite this enhancing dimensions. Additionally, the more static air, could be damaging to some of the electronic components and this remains to be tested. At high altitude where pressure is low, the balance on the flow lines could be altered using the ability to adjust flow rates to overcome pressure drop. We note that the

box is quite sturdy and internal parts are well secured, thus in places were transportation uphill cannot take place by helicopter or plane, the box could well be transported by yak oxen or sledges. At coastal foggy field sites, one could experience riming of the box and one should consider how to best remove such from the outside of the box. We recommend as always when embarking on field work to do proper testing in similar conditions prior to deployment.

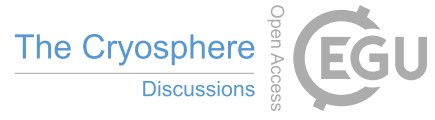

## 5 Conclusions

We have developed a Light Weight In Situ Analysis (LISA) portable CFA system and proved that it can work in the deep field at freezing temperatures (-20°C), to constrain snow pack peroxide, conductivity and, if density is also known, accumulation. The field work in 2017 shows that the box can work outside, and the 2019 field work proofs that it can be operated also by

scientists with limited or no prior CFA experience. Thus with this first prototype of the LISA box, we have a Portable, Practical and Proific prototype.

With this portable CFA system we have reconstructed accumulation in Northern central Greenland an area where only few previous constrains exist and thus added for 7 sites recent accumulation constrains that can be used as ground truth for satellite or radar reconstructions of accumulation or to assess precipitation in models covering the central Northern Greenland. We

found an increase in accumulation east of the divide compared to previous accumulation estimates, which we explain by the positive AO and NAO in the studied period. Similar could also be invoked to explain higher accumulation at the NEEM site west of the ice divide than those previously reported. We note, however, that close to the ice core drilling site EastGRIP local topography could also explain part of the difference to earlier accumulation estimates.

With its light weight and easy use, we expect the LISA system can be used also in smaller scale polar field operations and may

add to the kind of data that is reconstructed from the field.

**Supplementary**

Figure S1; The electrical circuit for the PID controlling the melt head temperature. Figure S2; the melt head design. Figure S3; Conductivity in top 100 cm from EGRIP 2017. Figure S4; Accumulation from North Greenland. Table S1; melt head specifications. Table S2; Accumulation results EGRIP 2017.


*Data availability*. The following files will be available free of charge from **www.Pangea.com** upon publication. Reconstructed accumulation and density for the 2 meter snow profiles (.xls), Peroxide and conductivity with depth for each of the 2 meter snow profiles (.txt).

*Author Contributions*. The manuscript was written through contributions of all authors. All authors have given approval to the final version of the manuscript. LLH, PV, HAK, NM and MS did the initial development of the Lisa box. LLH, HK and PV collected the samples during the field season 2017 and made initial testing of the box. HK, MS collected and analysed the snow samples on the LISA box in 2017. ZY, IK, MH, SK and JF collected samples from Northern Greenland sites as part of the NGT traverse during the field season 2019. ZY and IK ran the samples on the Lisa box at the EastGRIP site in 2019, while

MH and JF analyzed the samples for densities. HK, and LLH made the annual layer counting and reconstructed the accumulation for 2019 and 2017 respectively and HK made further analysis of the accumulation.





*Acknowledgements*. We would like to thank Mayu Lund, Romain Duphil, Jens Christian Hillerup, Angelika Humbert and Ole Zeising for assisting in the field. The Alfred-Wegener-Institut (AWI) operates the research planes Polar 5 and Polar 6. AWI funded the flight campaign ExNGT_PpRES_2019 with Polar 5 to visit the sites NEEM, B16, B19, B20, B22 and B27/28. We thank Daniel Steinhage (AWI) and the Basler crew for their support. We acknowledge EastGRIP for hosting us. EGRIP is directed and organized by the Center of Ice and Climate at the Niels Bohr Institute. It is supported by funding agencies and institutions in Denmark (A. P. Møller Foundation, University of Copenhagen), USA (US National Science Foundation, Office of Polar Programs), Germany (Alfred Wegener Institute, Helmholtz Centre for Polar and Marine Research), Japan (National Institute of Polar Research and Arctic Challenge for Sustainability), Norway (University of Bergen and Bergen Research Foundation), Switzerland (Swiss National Science Foundation), France (French Polar Institute Paul-Emile Victor, Institute for Geosciences and Environmental research) and China (Chinese Academy of Sciences and Beijing Normal University). The research leading to these results has received funding from the European Research Council under the European Community's Seventh Framework Programme (FP7/2007-2013) / ERC grant agreement 610055 as part of the ice2ice project and from the European Union's Horizon 2020 research and innovation programme under grant agreement No 820970 as part of the TiPES project and is TiPES publication #49. Additional support was received from the Villum Investigator Project IceFlow (no. 16572).

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
