# Peer review of "A portable Lightweight In Situ Analysis (LISA) box for ice and snow analysis"

_The Cryosphere, 2021_

## Author Response (AR1)

The authors would like to thank the reviewer for the valuable comments which will help to improve the quality of the manuscript. Kindly find below in red our response point-by-point to the reviewers' comments and suggestions.

**Comment on tc-2021-51**
Anonymous Referee #1
Referee comment on "A portable Lightweight In Situ Analysis (LISA) box for ice and snow analysis" by Helle Astrid Kjær et al., The Cryosphere Discuss.,
https://doi.org/10.5194/tc-2021-51-RC1, 2021

**Summary**
This paper presents a small, transportable, in-field continuous flow analysis (CFA) instrument, LISA. Choosing to measure just a few components that should well-record annual layers in Greenland, the authors aim to present improved annual accumulation records by considering spatial variability.
This new instrument is a novel contribution in that it takes CFA methodology, and creates a transportable unit easily operable in-field. CFA is usually part of more complex, permanent (or at least longer-term when rarely in-field) set-ups. The application of LISA is instead to be moved relatively quickly around several sites, geared towards rapidly obtained, possibly high-spatial resolution, annual accumulation records, assessing spatial heterogeneity.
As a first introduction to the new technology, this is a good contribution with potential for wider application. More accurate assessments of accumulation, and hence surface mass balance, contributes to improved scientific knowledge.
Based on these factors, the manuscript is recommended for publication, however this should be following the revisions as below.

**Major scientific comments**
The comparison of accumulation rates found here to those of other publications is not very robust and is somewhat unclear in presentation. This is because the other studies offer such a wide variation in timescales covered, many of which do not overlap with the years measured in this study. As Table 1 lays them out, it is too easy to scan across and think them more comparable than they are. Can you please either make this clearer by extending the table to include the dated age range of each average value presented in the others column, or remove this column from the table and include these values only in reference to the discussion in-text?

We have compared with the available accumulation reconstructions from snow and firn core that we are aware off. Unfortunately, not much exists with overlapping time periods. Thus we have included in our analysis all available. Regarding Table 1, While the information was already in the header it has been updated with also the exact time periods of the referenced studies to avoid any confusion about the time periods must often not overlapping between what we have analysed here and what is from the referenced studies. Further when referring to table 1 in the main text we have added the following sentence ". Note that only rarely do the estimates overlap in time."

Related to above, section 4.1 – The sentence on Clausius-Clapeyron should be removed. It is not possible to make a significant conclusion either way on this when comparing these very short, modern accumulation records of this study to other studies of average accumulation over assorted different time periods.
We have removed said sentence

Again on the above but in the conclusions section – these accumulation comparisons shouldn't become one of the main conclusions from the paper. Especially for NEEM, where you actually only have a record for one year of accumulation but start to invoke a positive AO and NOA to explain differences compared to longer average records. Perhaps with further records and more investigation this will be true, but there is not enough to say that here. Please re-focus the conclusions to the successes of the LISA method and application.

We have restructured the latter part of section 4.1 and added a sentence to emphasize the speculative nature of our AO/NAO comparison. The ssction on AO/NAO now starts with the following statement

*"In the following we speculate on AO and NAO impact on our records compared to previous estimates, we highlight that this is indeed speculative and that longer records are needed to firmly conclude on the impact of AO/NAO on accumulation."*

Further a few words have been altered from definite such as "can explain" to more soft words such as "could explain".

In the conclusion we have rewritten to not mention specifically AO/NAO; "*We found an increase in accumulation east of the divide compared to previous accumulation estimates and a higher accumulation at the NEEM site west of the ice divide than those previously reported, which we explain in part by the compared records not always overlapping in time and by natural variability in accumulation between years.*"

**Minor scientific comments**
Is it possible to rapidly stop core-melt in case of measurement issues to prevent loss of records?

As the current setup stands this is not possible to do in a "nice" way. That is you can remove the sample at any time and thus stop the analysis. One removes it by sliding up the black frame, and use a flat piece of metal or similar to lift the sample of the melt head. This will cause some loss of samples (¨1-2 cm), luckily in case of snow pits, there is plenty for new sample covering the same depth interval available, but then of cause shifted few centimeters from the original lost sample.

In section 5 we have added "In case of faulty analysis or blockages in the flow line, the sample and full frame can be lifted off the melt head, and restarted again once the issue is fixed. However, it will result in some loss of samples (1-3 cm). A benefit of moving the CFA to the field with this instrument is the option to obtaining new sample, and re-analyse. Something that is not possible if doing the analysis in the home laboratories. "

Or, if smaller sub-sampling from the waste-line is desired? How is this done?
We have added to section 5 *"Further discrete subsampling from either the melt head wasteline or an additional internal line is a possibility, such discrete samples may be useful for later analysis in home laboratories of proxies requiring more advanced setup such as ICP-MS or IC."*

From the photo in Figure 2 I cannot see any way in at the base of the liner to lift the core off the melt head for example. But maybe it is just not visible? Is this the cause of the gaps in the conductivity measurements in Figure 4?

Following on from this, can some explanation be given for Figure 4 for missing portions of data?

The small sections of missing data in Figure 4 (and 5) relates to the fact that most of the sites each of the two meter records are assembled from two snow samples each one meter. Due to smoothing in the system and response time of measurement systems, a little sample is lost at start and end of melting, and thus cause a small gap in the records, where the two snow cores are assembled. This could have been counteracted, by taking cores shifted slightly horizontally and then overlappig vertically. Unfortunately this was not done in 2019.
In figure 5 -S5 (red line) were melted by stacking the cores, thus avoiding the ramp up to sample.

We have added in section 4 the following "*The hydrogen peroxide results obtained from the LISA box in the 2019 field season are shown in Figure 5. Samples were analysed one meter at the time, resulting in small amounts of missing data at start and end of each sample as a result of the response time in the CFA system, the exception being for site S5 (red line), where the deeper sample (1-2 m) was added directly after the top one meter was analysed, thus avoiding the response time issue*."

Could an estimate of cost for such a LISA set-up be given in the supplementary information? A benefit of such light-weight, transportable systems is often that they are also lower-cost…is this one?

We will refrain from giving the exact cost of the LISA box in the manuscript as a lot of components used were from earlier CFA setups and thus today's prices may be different. In addition, some components were custom build at NBI/AWI (eg. melt head, F box, and the heat system for the box), and the cost of the manpower used is hard to assess. But overall I would assume cost of purchable items on the order of 27000 euro.

The reporting of melt speed measurement is a little inconsistent – in section 2.1 it is defined as the time taken to measure 3cm of ice. In section 3.1 it is 'Melt speed was registered by measuring the distance from the top of liner down to the top of the snow surface inside the liner about every three cm', which repeats from before but doesn't mention timing the change, and then in section 3.2 it is 'Depth registration was done every approximately 100 seconds and melt speed varied between 2.3 cm/min and 3 cm/min', so this was a slightly different method? Perhaps the method overview could simply state that it is measured by height change of the top of the snow over time, not defining 3cm which wasn't the case for the 2019 campaign, and then in the other sections report in more detail the exact methods. Also decide on whether to report the observed rates for 2017 or remove them for 2019 for consistency

We have stated in section 2.1 we now state just "*The melt speed in the portable CFA developed here is simply registered by simultaneously determining the amount of core left above the melt head with a ruler and the time*" and thus omit mentioning 3 cm.

In section 3.1 it now reads *"The amount of core left above the melt head and time was registered for about every 3 cm melted. "*

And in section 3.2 ; *"The amount of core left above the melt head was registered every approximately 100 seconds and melt speed varied between 2.3 cm/min and 3 cm/min. "*

**Technical comments**
Text varies between H2O2 and H2O2. It would be better to use 'hydrogen peroxide' as opposed to 'peroxide' in text since peroxide is a full class of chemicals rather than H2O2.
The suggested changes have been implemented

Inconsistent use of American/English spelling in some cases (analysed/analyzed).

P1L10: Suggest either 'Spend enormous amounts' or 'There are enormous costs involved in transporting…'
The suggested changes have been implemented

P1L23: SMB are made
The suggested changes have been implemented

P1L24: but also rely….from past periods
The suggested changes have been implemented

P1L28: fine cores are cheaper
The suggested changes have been implemented

P2L7,16: CFA is defined twice, and is again written in full P3L6, P5L11…I didn't check any further if perhaps you could?

The suggested changes have been implemented

P8L15 oC

The suggested changes have been implemented

P12L15 14[th]
The suggested changes have been implemented

Section 4.1: 'We note' is repeated four times here and is not needed.
3 out of the four were removed
P13L23: highlight that
The suggested changes have been implemented

P13L30: While we
The suggested changes have been implemented

P13L31: section
The suggested changes have been implemented

P14L20: remove 'as stands'
The suggested changes have been implemented

P15L5: remove capitalisation, and should be prolific?
The suggested changes have been implemented

P15L8: constrains repeats

The suggested changes have been implemented

**Comment on tc-2021-51**
Anonymous Referee #2
Referee comment on "A portable Lightweight In Situ Analysis (LISA) box for ice and snow analysis" by Helle Astrid Kjær et al., The Cryosphere Discuss., https://doi.org/10.5194/tc-2021-51-RC2, 2021

The paper "A portable Lightweight In Situ Analysis (LISA) box for ice and snow analysis" by Helle Astrid Kjær and coauthors reports the description, application and preliminary results of a new simplified portable CFA apparatus. The system, as shown in the paper, is able to continuously melt firn cores and to measure a bunch of parameters in the meltwater stream, namely conductivity and hydrogen peroxide. The system can be improved with additional analysis lines (nitrate, dust, ammonium, calcium etc) and it fits in an insulated box. Since this system is easily transportable it could be of interest for the ice core community but it can surely be improved in the future. I think that the paper is suitable for publication in TC, after that the following issues will be properly addressed.

Major comments

The main flaw of the paper is that the discussion about accumulation rate is a bit misleading and should be clearly assessed that it is just a speculative discussion. In fact, the recent accumulation rates at several sites were compared with very long accumulation rate histories and the authors drew some conclusions from this comparison. The text

dealing with the accumulation rate discussion with respect to previous records needs to be made clearer, pointing out when there is an overlap among the records and when there is a lack of overlap.

We have adjusted the manuscript accordingly, updated table 1, section 4.1 and the conclusion-see details in the response to reviewer 1 above and in the edited draft.

A point of weakness of this new instrumentation is the method used for the depth assignment of the analyzed ice sections. The authors propose different solutions to solve this problem but they need to find a robust and portable solution to be added to LISA. In the text is not mentioned if a weight on the core section is used to help the melting speed being constant. If not used, I would suggest to try this solution in order to have a relatively constant melt rate, more independent from the amount of ice left during the melting procedure, as already described in several papers (i.e. Severi et al., 2015, Anal. Chem).

We note that despite the uncertainties related to the method we use for registering depth at the moment, we do get comparable results to previous accumulation estimates. Thus despite other more precise options as described already in the paper, we will leave this for future improvements and a potential future paper.
In addition, our melt speed seem not to vary very much coming to the end of a sample. This is because we melt so relatively fast, that the temperature of the melt head end up controlling the speed more than the weight above. Further the samples we have investigated here were rather fragile and would have been rushed if adding weight on top.
.
*"In CFA systems the melt speed is often stabilized by adding a weight on top of the sample, to hinder a slow down of melt speed towards the end of melting (Bigler et al., 2011; Severi et al., 2015). For the analysis done here we have worked on snow samples, that would be compacted if such a weight was added and further we did not with the melt head temperature settings and the resolution of the depth registered observe any notable slow down in the melt speed. However, for more solid firn and snow samples, adding such a weight is an option also for the portable CFA. In case a firn core was analysed, another option is to stack the following core, once ~20 cm is left of the previous. This in addition to stabilizing the melt speed avoids the small amount of sample lost when for each subsample starting and ending measurements."*

Minor comments and typos
Abstract: I would recommend to remove the reference to figure 1 in the abstract. The abstract should be self-consistent.

The suggested changes have been implemented

Page 1 line 24 and 26 and several other times along the manuscript: missing spaces before brackets.
The suggested changes have been implemented

Figure 3: the flow rates reported in figure 3 are not consistent with the text. Is the flow rate of the melted sample 3.0 or 5.0 mL/min? And the same for the reagents. Please, correct the figure or the text.

Should be 3 mL/min text corrected. However, actually the flows can be varied by adjusting the pump tubing used or the flow speed of the pump. A line on this has been added at the end of the section "2.4.1 debubbler"

Page 7 line 18: Was the H2O2 reagent kept frozen or just refrigerated?
It was kept frozen until just prior use

Page 7 line 27: remove "both"
The suggested changes have been implemented

Page 8 line 15: change to °C

The suggested changes have been implemented

Table 1: use superscript for kg/m3. The w.e accumulation should be expressed as cm yr-1
The suggested changes have been implemented

Page 12 line 6: "by assuming that"
The suggested changes have been implemented

Page 12 line 18-19: this sentence is not clear. I can guess its meaning but it should be rephrased.
We have elaborated and the sentence now reads "3) that the seasonal timing of conductivity deposition over Greenland varies between sites. A spatial variability in the annual peak conductivity deposition is not unexpected as the conductivity in snow and ice, despite in Greenland being mostly driven by $H^+$ can be highly influenced also by forest fire acids and or salt content, which seasonal peaks may also vary from site to site.
"
Page 13 line 21: do you mean "simultaneously"?
Yes. The suggested changes have been implemented

Page 13 line 22: change to "highlight that"

The suggested changes have been implemented

---

## Author Response (AR2)

Dear Joel Savarino and type setting team

Please find enclosed the final version of our manuscript entitled **"A portable Lightweight In Situ Analysis (LISA) box for ice and snow analysis"** as accepted by the editor.

Only modification is to the data availability section. The data is uploaded to Pangea as one xls file only.

However Pangea warns " Depending on the extent and complexity of your data submission the editorial process and minting of DOI names might therefore take up to several months. Due to the high volume of submissions, it is unfortunately not possible to answer questions about the current processing status."

Thus we wonder if an update to tour publication can be made, once the data is finally published on Pangea.com and have received also a DOI and have for now stuck with the following data statement
*"The following files will be available free of charge from www.Pangea.com upon publication. Reconstructed accumulation and density for the 2 meter snow profiles, as well as hydrogen peroxide and conductivity with depth and age (.xls)."*
as the only modification to the accepted version.

Yours faithfully,

18. JUNI 2021

**HELLE ASTRID KJÆR**

ASSISTANT PROFESSOR

CENTRE FOR ICE AND CLIMATE

NIELS BOHR INSTITUTE

TAGENSVEJ 16

DK-2200 COPENHAGEN Ø

PHONE  +45 50528004

EMAIL   HELLEK@FYS.KU.DK

WWW    http://iceandclimate.dk